# Quasiperiodic Quadrupole Insulators

Raul Liquito[1], Miguel Gonçalves[1,2], Eduardo V. Castro[1,3]

[1]*Centro de Física da Universidade do Porto, Departamento de Física e Astronomia,*
*Faculdade de Ciências, Universidade do Porto, 4169-007 Porto, Portugal*
[2]*CeFEMA, Instituto Superior Técnico, Universidade de Lisboa,*
*Av. Rovisco Pais, 1049-001 Lisboa, Portugal and*
[3]*Beijing Computational Science and Research Center, Beijing 100084, China*

Higher-order topological insulators are an intriguing new family of topological states that host lower-dimensional boundary states. Concurrently, quasi-periodic systems have garnered significant interest due to their complex localization and topological properties. In this work we study the impact of quasi-periodic modulations on the paradigmatic Benalcazar-Bernevig-Hughes model, which hosts topological insulating phases with zero-energy corner modes. We find that the topological properties are not only robust to the quasi-periodic modulation, but can even be enriched. In particular, we unveil the first instance of a quasi-periodic induced second-order topological insulating phase. Furthermore, in contrast with disorder, we find that quasi-periodic modulations can induce multiple reentrant topological transitions, showing an intricate sequence of localization properties. Our results open a promising avenue for exploring the rich interplay between higher-order topology and quasi-periodicity.

## I. INTRODUCTION

Higher order topological insulators (HOTIs) have been established as an intriguing novel topological state of matter. Unlike the more conventional first order topological insulators, for which the bulk-boundary correspondence guarantees the existence of $D-1$ dimensional boundary states, HOTIs are characterized by boundary states with lower dimensionality. In particular, a $m^{\text{th}}$ order topological insulator has $D-m$ dimensional gapless boundary modes for a $D$-dimensional system [1]. Since their proposal several higher order topological phases have been characterized and observed in insulators [1–7], semimetals [8], superconductors [9, 10] and fractal lattices [11]. Just like the well known first order topological systems, HOTIs are symmetry protected while displaying more complex gap-edge dynamics. Considerable efforts have been made to unveil the symmetries behind the protection of these topological phases and to understand their robustness against weak disorder. HOTIs have been experimentally realized in phononic metamaterials [12], electric circuits [13, 14] and even real solid-state materials [15–17]. Disordered HOTIs have been studied [18–21] and a classical analog has been experimentally realized in electric circuits [14].

A different class of systems that break translational invariance, where more exotic localization properties can occur, are quasi-periodic systems. Contrary to disordered systems, extended, localized and critical multifractal phases can arise even in one dimension (1D) [22–31]. In higher dimensions, these systems have received considerable attention, including on the interplay between moiré physics and localization [32–39]. Quasi-periodic systems are also known to display intrinsic topological

properties and associated edge physics characterized by topological invariants defined in higher dimensions [40–45]. Systems with quasi-periodic modulations can be realized in widely different platforms, including optical lattices [46–55], photonic [40, 56–61] and phononic [62–67] metamaterials, and more recently using moiré materials [68–70]. The impact of quasi-periodic modulations on parent first-order topological systems has also been previously studied [71–75], and was found to give rise to interesting topological phases with richer localization properties than in the disordered cases. The effects of quasi-periodicity remain unstudied in higher order topological system, to our knowledge.

In this work we study a quasi-periodic (QP) chiral quadrupole insulator presenting a full characterization of topological, spectral and localization properties. The main results are shown in Fig. 1, where we plot the quadrupole moment $q_{xy}$ on [1(a)] and the bulk energy gap [1(b)] in the plane of intracell hoppings strength, $\gamma$, and QP modulation, $W$. As seen in Fig. 1(a), the clean limit quadrupole insulator (QI) and the trivial insulator (T) are robust to QP modulations. Starting from the clean limit trivial regime ($\gamma > 1$), these modulations may induce a topological phase transition (TPT) into a QI phase adiabatically connected to the clean limit QI for $\gamma \gtrsim 1$. For the topological case ($\gamma < 1$), the QP modulations eventually induce a TPT into a gapless critical metal. Further increasing the strength of the QP modulation leads to a novel reentrant topological regime which we entitle quasi-periodic QI (QPQI). The QPQI phase display similar boundary signatures as their clean limit counterparts, such as four fold zero energy corner modes that give rise to fractional corner charges. However, in this regime an intricate interplay between QP induced edge states and corner modes emerges, which can lead

to edge-corner hybridization. This is a unique feature of the QPQI phase. Overall, the observed TPT between the various regimes occur in gap closing/opening as seen in Fig. 1(b).

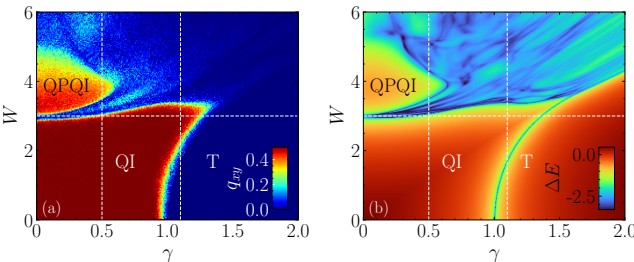

FIG. 1. Phase diagrams in the plane $(\gamma, W)$. (a) The quadrupole moment was computed for a system size $L = 13$ with $\theta_i = \pi$ and 100 averages over phase shifts were realized. (b) Spectral gap $(\Delta E)$ in the $(\gamma, W)$ plane for a system size $L = 34$ with $\theta_i = 0$ and 50 realizations of phase shifts. A minimum value of the spectral gap was chosen $(\Delta E < 10^{-3})$ to highlight all the gapped and gapless phases appearing in the phase diagram. The dotted white lines are constant $\gamma$ and $W$ cuts for $\gamma = 0.5, 1.1$ and $W = 3$ respectively.

This paper is organized as follows: in Sec. II we introduce the quasi-periodic Benalcazar-Bernevig-Hughes model and present the methods used to numerically study the diverse phases that emerge under QP modulations; in Sec. III we display and discuss our results in a organized fashion; in Sec. IV we discuss and reinforce our main conclusions; in Appendix A we show calculations of the DOS via KPM for each relevant observed phases and in Appendix. B we perform the same analysis done throughout this work but for a constant QP modulation strength $(W)$ cut (varying the inter unit cell hopping $\gamma$); in Appendix C some remarks are made regarding phase twists; and in Appendix D we expand on the results of phase shifts dependence in the QPQI phase.

## II. MODEL AND METHODS

### 1. Model

We consider the BBH model [1] with a QP modulation defined by the following Hamiltonian:

$$\mathcal{H} = \sum_{\mathbf{R}} \left( \mathbf{\Psi}_{\mathbf{R}}^{\dagger} \Gamma_{\mathbf{R}} \mathbf{\Psi}_{\mathbf{R}} + \sum_{i \in \{x,y\}} \mathbf{\Psi}_{\mathbf{R}+\hat{\mathbf{e}}_i}^{\dagger} \Lambda_i \mathbf{\Psi}_{\mathbf{R}} \right), \quad (1)$$

where $\mathbf{\Psi}_{\mathbf{R}}^{\dagger} = \left( c_{\mathbf{k},1}^{\dagger} \; c_{\mathbf{k},2}^{\dagger} \, , c_{\mathbf{k},3}^{\dagger}, c_{\mathbf{k},4}^{\dagger} \right)$ and $c_{\mathbf{R},\alpha}^{\dagger}$ creates an electron on the orbital $\alpha$ at site $\mathbf{R} = (x, y)$. The model is defined by the hopping matrices:

$$\Gamma_{\mathbf{R}} = \delta\sigma_z \otimes \sigma_0 + \gamma_{\mathbf{R}} \left( \sigma_x \otimes \sigma_0 - \sigma_y \otimes \sigma_y \right)$$

$$\Lambda_x = \frac{\lambda}{2} \left( \sigma_x \otimes \sigma_0 - \sigma_y \otimes \sigma_y \right) \quad (2)$$

$$\Lambda_y = i\frac{\lambda}{2}\sigma_y \otimes \left( \sigma_x + \sigma_z \right),$$

where $\{\sigma_i\}$ are the set of Pauli matrices. Throughout this paper we set $\lambda = 1$ such that $\gamma$ is displayed in units of $\lambda$. The QP modulation is introduced on the inter UC hopping's,

$$\gamma_{\mathbf{R}} = \gamma + \frac{W}{2} \cos\left(2\pi\beta x + \phi_x\right)\cos\left(2\pi\beta y + \phi_y\right), \quad (3)$$

where $\beta$ is an irrational number. This choice of QP modulation ensures that chiral symmetry remains preserved for any QP modulation strength $(W)$. Furthermore, this model also preserves time reversal and charge conjugation symmetries. For a generic phase shift $\boldsymbol{\phi} = (\phi_x, \phi_y)$, the QP modulation breaks the mirror and $C_4$ crystalline symmetries. Therefore, this symmetry is broken for the finite-size systems that we study. Nonetheless, it should be recovered in the thermodynamic limit, when there should be no $\boldsymbol{\phi}$ dependence of the bulk properties.

We carried out numerical simulations for finite systems with $L_x = L_y = L$ (with $L$ the number of unit cells in each direction) and periodic/twisted boundary conditions. In order to avoid boundary defects the system sizes were chosen to be $L = F_n$, where $F_n$ is the $n$-th order Fibonacci number. In Eq. (3), $\beta$ was taken as a rational approximant of the golden ratio $\beta \to \beta_n = F_{n+1}/F_n$. This choice ensures that the system's unit cell is of size $L$, guaranteeing that the system remains incommensurate as $L$ increases. Furthermore, throughout this work we average over phase shifts sampled from a random uniform distribution $(\phi_x, \phi_y \in [0, 2\pi[)$. Regarding twisted boundary conditions, for some quantities such as the IPR we also take simultaneous averages over phase twists $(\boldsymbol{\theta} = \theta_x, \theta_y)$ while for spectral gap dependent results we fixed a twist of 0 or $\pi$ depending if the system size is even or odd respectively (see Appendix C). To apply phase twists, the boundaries are periodically closed (as for periodic boundary conditions), but with an additional twist, such that:

$$\psi_\alpha(\boldsymbol{R} + L\mathbf{a}_i) = e^{i\theta_i}\psi_\alpha(\boldsymbol{R}), \, i = x, y \quad (4)$$

where $\psi_\alpha(\boldsymbol{R}) = c_{\mathbf{R},\alpha}^{\dagger} |0\rangle$. As the system approaches the thermodynamic limit, any bulk dependence on phase twists should vanish.

In the clean limit $(W = 0)$ the system preserves translational invariance and we re-obtain the BBH model [1], described by the following Bloch Hamiltonian:

$$\mathcal{H}_{\mathbf{k}} = \left[\gamma + \lambda\cos(k_x)\right]\Gamma_4 + \lambda\sin(k_x)\Gamma_3 + \\ + \left[\gamma + \lambda\cos(k_y)\right]\Gamma_2 + \lambda\sin(k_y)\Gamma_1 + \delta\Gamma_0 \quad (5)$$

where $\Gamma$ are $4 \times 4$ matrices ($\Gamma_0 = \tau_3 \otimes \sigma_0$, $\Gamma_k = -\tau_2 \otimes \sigma_k$, $\Gamma_4 = \tau_1 \otimes \sigma_0$) that define the internal degrees of freedom within a unit cell.

For $|\gamma| < 1 (|\gamma| > 1)$ the system is in a topological (trivial) phase defined by $q_{xy} = 0.5$ ($q_{xy} = 0$). As for the topological phase, the system displays fourfold zero energy corner modes that give rise to fractional corner charges $Q_i = \pm 0.5$. Since this quadrupole insulating phase is a second order topological insulator (SOTI), the boundary is a first order topological insulator. In fact, from the boundary perspective the corner modes appear as localized edge states. In this manner, we can characterize the topological phase by calculating the boundary polarizations with the nested Wilson loop approach introduced and discussed in Refs. [1, 76]. It was shown that in the presence of quantizing crystalline symmetries ($C_4$, and reflection along $x$ and $y$, $M_x$ and $M_y$), the quadrupole insulating phase obeys $q_{xy} = p_x = p_y = 0.5$, and that anti-commuting mirror symmetries are needed to have a Wannier gap. For any given finite value of $W$ these crystalline symmetries are broken for a finite system with generic $\phi$, however, it has recently been shown that the quadrupole moment is equally quantized by chiral symmetry [77, 78] and that a new set of $\mathbb{Z}$ topological invariants arise, known as the multipole chiral numbers (MCNs) [79]. Although this new characterization is shown for systems falling in the AIII symmetry class, it should be valid for any of the chiral symmetric classes (AIII, BDI, CII), since BDI and CII classes display more symmetry than the AIII class. These chiral symmetric quadrupole phases are topological and display the same properties as their crystalline counterparts, such has zero energy corner modes, fractional corner charges, quantized boundary polarizations and bulk quadrupole moments.

## 2. Methods

*Non-trivial topology* – To compute $q_{xy}$ for a disordered system the multipole operators are considered [80, 81], which are generalizations of the Resta's formula [82],

$$q_{xy} = \left[ \frac{1}{2\pi} \text{Im} \log \langle \Psi_0 | e^{2\pi i \sum_{\mathbf{r}} \hat{q}_{xy}(\mathbf{r})} | \Psi_0 \rangle - q_{xy}^0 \right] \mod 1, \tag{6}$$

where $\hat{q}_{xy}(\mathbf{r}) = \frac{xy\hat{n}(\mathbf{R})}{L_x L_y}$, $|\Psi_0\rangle$ is the ground state of the system and $q_{xy}^0$ the background positive charge contribution for the quadrupole moment. To study boundary topology, we also compute the boundary polarizations making use of the effective boundary Hamiltonian defined through $\mathcal{H}_{\text{Bound}} = G_N^{-1}(E=0)$, where $G_N(E)$ is the boundary Green's function [83]. To obtain $G_N(E=0)$, we take a transfer matrix approach and divide the system in 1D strips that are connected by hopping matrices $V_n$. The boundary Green's function is then obtained with the

following Dyson equation:

$$G_n(E) = \left( E\mathcal{I} - h_n - V_{n-1} G_{n-1} V_{n-1}^\dagger \right)^{-1} \tag{7}$$

where $h_n$ is the Hamiltonian of the $n^{\text{th}}$ strip. To compute the boundary polarizations, we resort to Resta's formula:

$$p_i = \left[ \frac{1}{2\pi} \text{Im} \log \langle \Psi_c | e^{2\pi i \sum_{\mathbf{r}_i} \hat{p}(x_i)} | \Psi_c \rangle - p_i^0 \right] \mod 1 \tag{8}$$

where $\hat{p}_i(\mathbf{r}) = \frac{x_i \hat{n}(x_i)}{L_x}$, and $|\Psi_c\rangle$ is the boundary ground state obtained with exact diagonalization of the boundary Hamiltonian. With these polarizations we can define a boundary invariant $P = 4 |p_x p_y|$ such that $P = 0$ for $q_{xy} = 0$ and $P = 1$ for $q_{xy} = 0.5$ as in Ref. [77][84]. We note that both $q_{xy}$ and $P$ are quantized for every configuration, even if the system is gapless. However, the average over phase shifts $\phi$ and phase twists $\theta$ can lead to non quantized $q_{xy}$ and $P$.

*Spectral Methods* - To study the spectral properties of our system we compute the spectral gap ($\Delta E$) via sparse diagonalization with shift invert and cross check the results with the density of states (DOS) at Fermi level $\rho(E = 0)$. The DOS is defined as $\frac{1}{L_x L_y} \sum_i \delta(E - E_i)$ and it was computed with an implementation of KPM as presented in Ref. [85] making use of the Jackson kernel. We also define the corner occupation probability,

$$P_{occ} = \sum_{\mathbf{R} \in \text{corner}} \sum_{\alpha=1}^{4} |\psi_n^\alpha(\mathbf{R})|^2, \tag{9}$$

where $\psi_n^\alpha(\mathbf{R})$ represents wavefunction amplitude in the $\alpha^{\text{th}}$ orbital at site $\mathbf{R}$ for the eigenstate with energy $E_n$. For the calculations that follow, we average $P_{occ}$ over the four eigenstates closest to the Fermi level obtained with Lanczos decomposition and shift invert. In a QI phase, zero energy corner modes occur and thus $P_{occ} \approx 1$. For a trivial phase, no corner modes exist and $P_{occ} < 1$ is expected. Since the corner states display a localization length dependent on $W$ and $\gamma$, we compute $P_{occ}$ over an $l \times l$ region which we assume to be the corner, i.e. we restrict the sum in Eq. (9) to that region. Therefore, $l$ is effectively an estimation of the localization length of the corner modes. Since the zero energies corner modes give rise to fractional corner charges, we also compute the corner charge $\bar{Q} = \sum_i |Q_i|$ as a function of $W$, obtained from $Q_i = \sum_{\mathbf{R} \in \text{corner}} \rho(\mathbf{R})$, where $\rho(\mathbf{R}) = 2 - \sum_{n \in \text{occ.}} \sum_{\alpha=1}^{4} |\psi_n^\alpha(\mathbf{R})|^2$ is the charge density. The first and second terms arise from the atomic positive charges and from the electronic density, respectively. Unlike the computations for $P_{occ}$, the system is partitioned into four quadrants, each of the size of a quarter of the system. Then, the corner charge is computed by

integrating $\rho(\mathbf{R})$ over each of the quadrants. By definition, the corner charge requires complete knowledge of the systems spectra, obtained by full diagonalization of the systems Hamiltonian, limiting the maximum system sizes to $L = 55$. In quadrupole insulating phases $\bar{Q} = 0.5$.

*Localization Methods* - To tackle localization properties several quantities are computed such as the inverse participation ratio (IPR), $k$-space inverse participation ratio (IPR$_k$), fractal dimension ($D_2$) and the localization length. The normalized localization length $\xi/M$ where $\xi$ is the localization length and $M$ the transversal system size, was obtained via transfer matrix method (TMM) for a choice of longitudinal size that yields an error $\epsilon \approx 1\%$. For localized states, $\xi/M \to 0$, for extended states $\xi/M \to \infty$ and for critical states $\xi/M$ remains constant.

The IPR [86] is defined as:

$$\mathrm{IPR} = \frac{1}{N} \sum_{\mathbf{R},\alpha} |\psi_\alpha(\mathbf{R})|^4 \qquad (10)$$

where $\psi_\alpha(\mathbf{R}) = \langle \mathbf{R}, \alpha | \psi \rangle$ for a normalized eigenstate $\langle \psi | \psi \rangle = 1$. From the scaling IPR $\propto L^{-D_2}$, we extract the fractal dimension $D_2$ by fitting the results for different system sizes. Thus for $D_2 = 2$ the eigenstates are extended, for $D_2 = 0$ the eigenstates are localized, while for $0 < D_2 < 1$ the eigenstates display critical character.

## III.   RESULTS

The phase diagram of the system is presented Fig. 1 on the $(\gamma, W)$ plane, where topological properties are shown in panel 1(a) and spectral properties in panel 1(b). In what follows, we study in detail the cuts indicated in Fig. 1, focusing on topological, spectral, localization and edge properties.

### A.   Topological Properties

*Starting Topological* - The results obtained by increasing $W$ starting at a topological phase are shown in Figs. 2(a) and 2(c). For $\gamma = 0.5$ and $W = 0$ the system is in a topological phase. The quadrupole moment and the boundary invariant are quantized respectively to 0.5 and 1 for each realization of phase shifts, remaining quantized with increasing $W$, demonstrating that the quadrupole phase is robust to quasi-periodic modulations. Eventually, at $W_c^{I \to II} \approx 3.05$ the gap closes [see panel 2(c)] and a TPT occurs, with the system entering a critical metal phase [II$a$, see inset of Fig. 2(c)], which we motivate below after analyzing the localization properties. Further increasing $W$, the gap reopens into a topological regime [II$b$ in panel 2(c)]. Focusing on the $P$ invariant (or

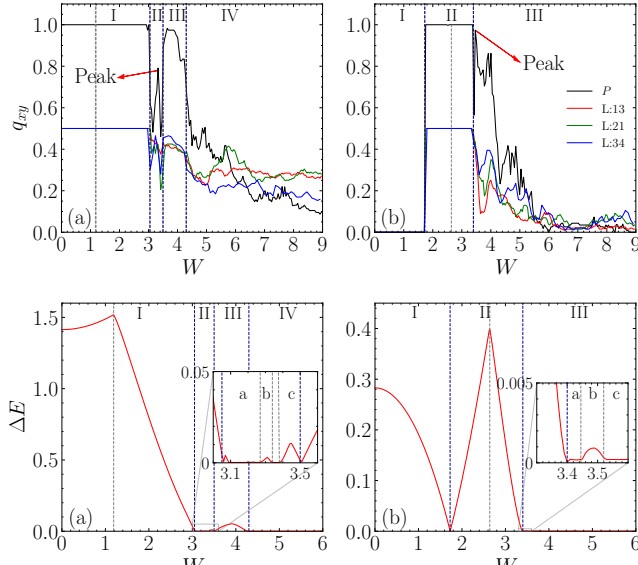

FIG. 2. (a-b) $q_{xy}$ and $P$ as a function of $W$. The $P$ invariant was computed for a system size of $L = 610$ with 115 averages over phase shifts. $q_{xy}$ was obtained via real space methods with 100 averages over phase shifts. For even $L$, $\theta_x = \theta_y = 0$; for odd $L$ $\theta_x = \theta_y = \pi$. (a) $\gamma = 0.5$. (b) $\gamma = 1.1$. (c-d) Spectral gap ($\Delta E$) as a function of $W$ computed for an even system size $L = 144 \implies \theta_i = 0$ and averaged over 50 phase shifts realizations. (c) $\gamma = 0.5$. (d) $\gamma = 1.1$.

the quadrupole moment $q_{xy}^{L=34}$), a sharp peak can be observed in this region. The fact that this peak approaches 1 (0.5 for $q_{xy}^{L=34}$) with increasing system size, while the value of $P$ (or $q_{xy}^{L=34}$) in II$a$ and II$c$ go to zero, hints that II$b$ is a quadrupole insulating phase. Furthermore, since this is a very narrow $W$ window and the gap is small ($\Delta E \approx 10^{-3}$) we attribute the lack of quantization to finite size effects. Increasing $W$ even more induces a trivial insulating phase (II$c$) before another quadrupole insulating regime is reached (III) that suffers from the same finite size effects as II$b$. In this manner, QP modulations can induce a novel type of SOTI phases which we entitle Quasi-Periodic Quadrupole Insulator (QPQI) phases. At high $W$, the gap closes (at $W_c^{III \to IV} \approx 4.3$) and the system reaches a gapless regime (IV) with a finite, although not quantized, quadrupole moment ($q_{xy} \neq 0$).

*Starting Trivial* - The results obtained by increasing $W$ starting at a trivial phase are shown in Figs. 2(b) and 2(d). For $\gamma = 1.1$ and $W = 0$ the system is in a gapped trivial phase, which is stable to the application of a QP modulation (I). Increasing $W$ induces a TPT into a gapped QI phase (II) with the gap closing and reopening at $W_c^{I \to II} \approx 1.73$. Furthermore, the gap reaches its maximum, with a level crossing between energy levels in the gap edge occurring at $W \approx 2.64$ (dotted grey line). Away from the QI phase, the gap closes ($W_c^{II \to III} \approx 3.4$) and reopens in IIIb with a sharp peak in $P$ that almost

quantizes to 1. Just as in the previous case ($\gamma = 0.5$), we attribute this lack of quantization to finite size effects and classify IIIb as QPQI phase. Eventually, at the high $W$ regime, the system undergoes a TPT into a gapless regime (IIIc).

We note that there are regions where there could be an even more intricate structure of gap and gapless regions with reentrant topological transitions (as unveiled by the complex behavior of $P$ and $q_{xy}$) which would only be possible to capture with much larger system sizes.

## B. Density of States

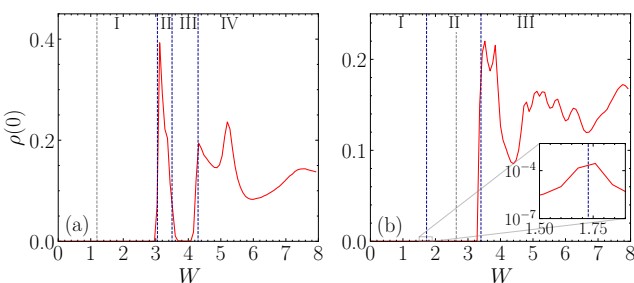

FIG. 3. The DOS at zero energy ($\rho(0)$) as a function of $W$. The following KPM parameters were used: $L = 987$ $M = 5000$ Chebyshev moments, and $R = 10$ stochastic traces. (a) $\gamma = 0.5$. (b) $\gamma = 1.1$.

In Fig. 3 we plot the DOS at $E = 0$ as a function of $W$ with the intent of verifying the results of the spectral gap presented in Fig. 2. In gapped regions $\rho(0) \approx 0$, while in gapless regions the system should have a finite DOS. All gapless phases (II and IV for $\gamma = 0.5$ and III for $\gamma = 1.1$) display a finite DOS at Fermi level. Furthermore, at the transition I $\to$ II ($\gamma = 1.1$) the gap closes and reopens at a critical QP potential strength ($W_c^{I \to II} \approx 1.73$), which is signaled by a small increase of $\rho(0)$ around it, as shown in the inset of Fig. 3(b). The Jackson kernel broadens the Dirac delta as a Gaussian with spread $\sigma \propto 1/M$, where $M$ is the number of Chebyshev moments. For this reason, it was not possible to reach enough resolution to capture the gapped IIb and IIc phases, that are filled by the tails of the Gaussians. The same occurs for $\gamma = 1.1$ in phase IIIb. These narrow phases are only expected to be resolved for larger system sizes and number of moments.

## C. Localization Properties

To complete the characterization of the phase diagram, we turn to the study of the localization properties. In Fig. 4 we show the normalized localization length ($\xi/M$) at the Fermi level ($E = 0$), obtained using the TMM. For the regions with larger gap, it is clear that $\xi/M$ scales

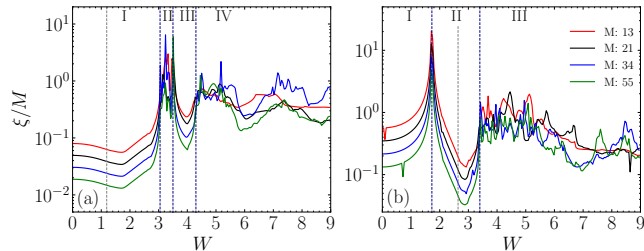

FIG. 4. TMM at Fermi level ($E = 0$) for several transversal system sizes ($M$). For odd $M$ we add a twits of $\theta_M = \pi$. The size of the longitudinal direction was chosen such that the relative error $\epsilon < 1\%$. (a) $\gamma = 0.5$. (b) $\gamma = 1.1$.

down to zero, since TMM captures localized evanescent wave solutions of the Schrödinger equation. At gapless regions, or regions with small gaps, the results are noisier and in some cases larger system sizes would be needed to understand whether $\xi/M$ converges with $M$, showing critical behavior, or decreases with $M$, signaling gapless localized states. This will become clearer from the eigenstate analysis based on exact diagonalization that we carry out below. Noteworthy, the topological phase transition occurring through gap closing and reopening as seen in Fig. 2(d) is well captured by a peak in $\xi/M$ in Fig. 4(b), which is expected to not decrease with $M$ precisely at the critical point.

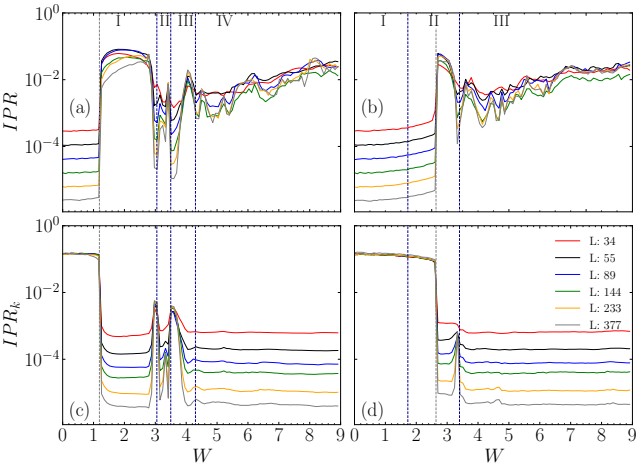

FIG. 5. (a-b) IPR and (c-d) $\text{IPR}_k$ for different system sizes obtained with exact diagonalization methods. The IPR and $\text{IPR}_k$ are averaged over 100 phase twists, shifts and over the first eight eigenstates with energies closest to $E = 0$. (a,c) $\gamma = 0.5$. (b,d) $\gamma = 1.1$.

To further characterize the localization properties of the eigenstates with energies closer to $E = 0$, we compute the IPR and the $\text{IPR}_k$. In the gapped phases, these states correspond to eigenstates at the gap edge that are not captured by the TMM at Fermi level. In Fig. 5, we plot the average IPR (a-b) and $\text{IPR}_k$ (c-d) of the eight

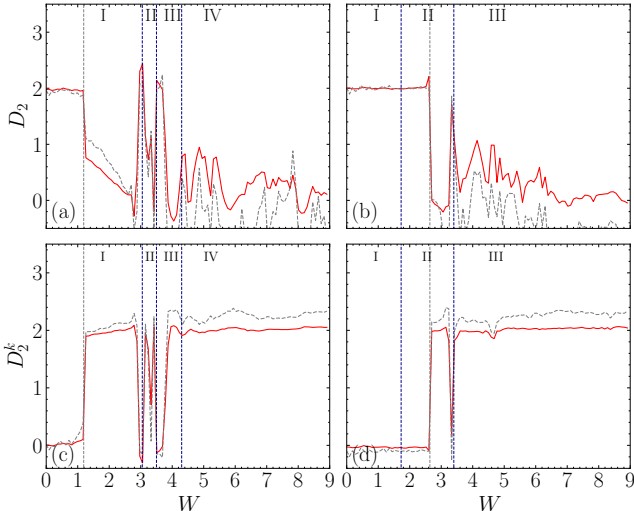

FIG. 6. $D_2$ (a,b) and $D_2^k$(c,d) for different system sizes obtained from the IPR and $I$IPR results respectively. (a,c) $\gamma = 0.5$. (b,d) $\gamma = 1.1$. The red curves were estimated making a fit in $\log - \log$ scale for all $L = \{34, 55, 89, 144, 233, 377\}$, while the dotted grey lines were obtained only considering the biggest three systems sizes $L = \{144, 233, 377\}$.

eigenstates closer to $E = 0$ as a function of $W$. We also plot the fractal dimensions $D_2$ and $D_2^k$ in Fig. 6 estimated, respectively, through the IPR and IPR$_k$.

*Starting Topological* - For $\gamma = 0.5$, in the gapped phase I, the eigenstates at the gap edge start off ballistic as seen in Fig. 6, with $D_2 \to 2$ and $D_2^k \to 0$. They undergo a ballistic→localized ($D_2 \to 0$ and $D_2^k \to 2$) transition at $W \approx 1.19$ (grey line) further ascertaining that a level crossing between gap edge states occurs. This transition is not captured by the TMM since the system remains gapped. At the TPT (I→II) the gap-edge states are ballistic, however, these states quickly turn critical as seen by $D_2 \approx 1$, indicating that the gapless critical metal regime is reached (IIa). The normalized localization length ($\xi/M$) obtained via TMM does not diverge at the transition, instead, it goes to a constant value [see Fig. 4(a)], in agreement with the critical metal regime. In the QPQI phase (IIb), the gap-edge states remain critical, localizing in the IIc phase. In these latter regimes TMM fails to capture such narrow dynamics, due to the lack of discretization in TMM calculations. At the transition from II→III, IPR, IPR$_k$ (Fig. 5) and $D_2$ (Fig. 6) indicate a ballistic regime with the the gap-edge states undergoing a ballistic-localized transitions halfway through phase III.

Phase IV is of difficult classification with any of the methods at our disposal. At the transition III→IV the IPR results suggest that the eigenstates are critical. After this transition, TMM and IPR results are noisy. However, considering the fractal dimension computed with the last three system sizes, $D_2$ approaches zero indicat-

ing that the system has reached an Anderson insulating phase ($D_2 \to 0$).

*Starting Trivial* - Starting at the trivial phase with $\gamma = 1.1$, the gap edge eigenstates are ballistic within phase I and remain ballistic even after the transition into phase II [$D_2 \to 2$ and $D_2^k \to 0$, see Fig. 6]. At the TPT (I→II), a gap closing and reopening topological transition with ballistic gap edge states occurs, as in the absence of quasi-periodicity. This explains why this particular gap closure is highly dependent on the choice of twists, for small system sizes [28]. As in the previous case, the ballistic regime is maintained until the level crossing between ballistic and localized gap edge levels occurs (grey line), where we observe $D_2 \to 0$. For higher $W$, slightly before the gap closing at $W_c^{II \to III} \approx 3.4$, gap edge states delocalize and then become critical at the transition ($D_2 \approx 1.2$).

The results for the final phase (III) are similar to phase IV of the previous cut, leading us to conclude that an Anderson insulating regime is reached.

## D. Opening the Boundaries

An important hallmark of topological quadrupole insulators is that quadrupole phases defined by $q_{xy} = 0.5$ have zero energy corner modes that give rise to fractional corner charges when the boundaries of the system are opened. For this reason we now study the system with open boundary conditions. In Fig. 7(a-b), we show the $P_{occ}$ as a function of $W$ and in Fig. 7(c-d) we show the averaged corner charge $\bar{Q} = \sum_i |Q_i|$ ($Q_i$ is the corner charge of each individual corner) as a function of $W$.

*Starting topological* - For $\gamma = 0.5$, it can be seen in Fig. 7(a) that $P_{occ}$ has perfect plateaus at $P_{occ} = 1$ for every studied $l$, indicating the existence of highly localized corner states ($\xi_l < 10$ where $\xi_l$ is the localization length of the corner modes). Similarly, the corner charge also quantizes at $\bar{Q} = 0.5$ throughout the topological phase, as shown in Fig. 7(c). In phase II, the critical metal (IIa) and trivial insulator (IIc) display a finite, but significantly smaller $P_{occ}$, that arises from contributions of edge or bulk states (below we will provide a detailed discussion on edge states). In the narrow topological regime observed (IIb), $P_{occ}$ sharply peaks, scaling closer to 1 as indicated by the arrow in Fig. 7(a). However, the peak does not properly quantize to 1 for the studied $l$ values, which we attribute to the large localization length of the corner modes, $\xi_l > 30$, due to the very small gap in this phase. A peak can also be observed in the corner charge for $L = 55$ [Fig. 7(c)], albeit smaller due to the considerably smaller system size. Regarding phase III, $P_{occ}$ has a noisy plateau that scales to 1 as $l$ increases, indicating again that the corner states display a larger localization length than the maximum considered corner ($l = 30$). The corner charge also scales to 0.5 with

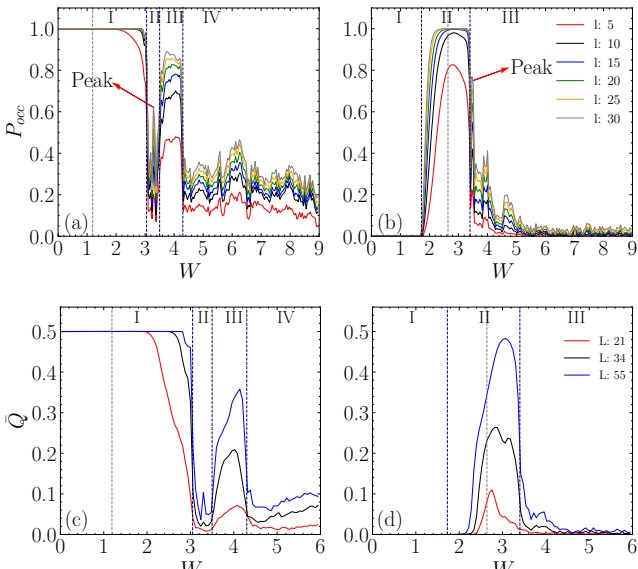

FIG. 7. (a-b) Corner occupation probability ($P_{occ}$) as a function of $W$ for the first two zero energy states. The results were obtained for a system with size $L = 377$ via Lanczos decomposition and shift invert and averaged over 200 phase shifts. (c-d) Corner charge ($\bar{Q}$) as a function of $W$ for several system sizes calculated via full diagonalization of the systems Hamiltonian and averaged over 100 phase shifts and over the four corners. A small $\delta = 10^{-5}$ was used in both cases to split the degeneracy of the four fold corner modes. (a,c) $\gamma = 0.5$. (b,d) $\gamma = 1.1$.

increasing size, further corroborating that III is indeed topological with corner modes characterized by a larger localization length. Finally, as expected, phase IV has a small $P_{occ}$. However, we also see a growth with increasing corner/system size. This increase is explained by the gapless nature of this phase. As $l$ increases, more and more bulk spectral weight fall within the considered $l \times l$ corner, leading to a visible increase in $P_{occ}$. Regarding the corner charge, although there are no quantized corner charges, for some realizations of phase shifts the electron density displays localized peaks around the average bulk electron density which leads to finite contributions to the corner charge. This occurs due to the breaking of chiral symmetry by the small $\delta$ introduced to lift the degeneracy of the corner modes. Overall the $P_{occ}$ and $\bar{Q}$ of each phase agree well with the topological phase diagram.

*Starting trivial* - For $\gamma = 1.1$ the trivial phases I and IIIc display $P_{occ} \approx 0$ as seen in Fig. 7(b). For phase II, the corner occupation probability, though initially suffers from finite size effects, eventually reaches 1. Its dependence on $l$ after the TPT (I → II) indicates a diverging localization length of the corner states as the critical point is approached. Deeper into the QI phase, the corner modes become more localized. Regarding IIIb, $P_{occ}$ has a small peak in this region as indicated by the

arrow in Fig. 7(b). Regarding the corner charge shown in Fig. 7(d), it displays scaling with increasing system size, however a plateau cannot be seen even for the biggest considered size $L = 55$. Moreover, the corner charge calculations do not have enough resolution to catch narrow topological regimes like the IIIb phase. We have checked for $P_{occ}$ in phase II that taking $l = 30$ for the linear size of the corner is not enough in a system with size $L = 377$ if we want to measure all the weight of the corner state. This is a clear indication that the corner states display a localization length $\xi_l > 30$. Therefore, simulating a system of size $L = 55$ [87], leads to the hybridization of the corner modes, which in turn leads to a non quantized corner charge.

*Properties of QPQI phases* - Until now a proper characterization of the QPQI phases has not been provided. In the paragraphs that follow, we aim to fully define these quasi-periodic induced regime. We start with their definition: a QPQI phase is a topological quadrupole insulating phase that is not adiabatically connected to any of the clean limit phases while displaying a quasi-periodic induced bulk-spectral gap (or is gapless but displays localized states at the Fermi level, as discussed in Appendix B).

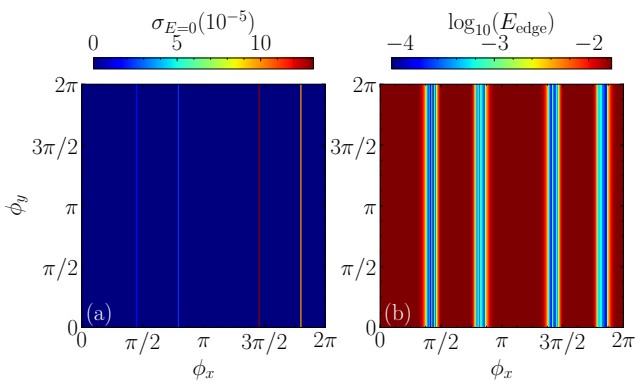

FIG. 8. (a) Standard deviation ($\sigma_{E=0}$) of the energies of the corner modes as a function of the phase shifts ($\phi_x, \phi_y$). (b) Logarithm of edge state energy ($\log_{10}(E_{edge})$) as a function of the phase shifts ($\phi_x, \phi_y$). Results obtained for $W = 3$, $\gamma = 0.5$ for a system size $L = 233$.

We have found that quasi-periodic induced gapped phases usually display edge states that disperse with the phase shifts. This phenomena is common in QP systems and can be observed in the Aubry-André model. As $\phi$ changes, the energy of the edge states can move closer or even cross the Fermi level. This phenomena is particularly relevant in topological regimes since the edge states can hybridize with zero energy topological modes, thus pulling them away from zero energy. We expect that when such crossing occurs the system behaves as in the trivial regime, displaying $P = 0$. Furthermore, $P_{occ}$ is also impacted since the corner modes are hybridized

with edge states that exhibit spectral weight outside of the corners.

In order to check the existence of edge states and study edge-corner hybridization we resort to exact diagonalization of the open boundary system Hamiltonian, calculating the six eigenvalues closest to the Fermi level. Assuming we choose a point of the phase diagram that resides in a SOTI phase, out of the six energies, four correspond to zero energy corner modes with the remaining two corresponding to the edge states with energy closest to the Fermi level. We then proceed to compute the standard deviation of the corner modes energy $\left( \sigma_{E=0} = \sqrt{\frac{1}{N} \sum_i E_i^2} \right)$ and the energy of the edge state closest to the Fermi level ($E_{edge}$). In this manner, when the edge states energy approach the Fermi level ($E_{edge} \to 0$) a peak should be seen in $\sigma_{E=0}$, indicating that the corner modes hybridized with the edge states.

Before tackling QPQI phases that display complex dynamics, we consider a simpler case. In Fig. 2(a), just before the TPT from I→II, a small dip can be seen in the $P$ invariant ($W \approx 3$ and $\gamma \approx 0.5$), suggesting that some dependence on the phase shifts might exist. In this manner, in Fig. 8(a) we show the density plot of $\sigma_{E=0}$ and in Fig. 8(b) the density plot for $E_{edge}$ in the ($\phi_x, \phi_y$) plane. Four lines of constant $\phi_x$ can be observed where $\sigma_{E=0} > 0$ [Fig. 8(a)] for $E_{edge} \to 0$ [Fig. 8(b)], indicating that the corner modes hybridized away from zero energy. Moreover, not all four corner modes hybridize with the edge states. In fact, only two of them move away from zero energy through hybridization, allowing us to distinguish two types of corner states. We attribute the term *inner corner modes* to the two corner modes that remain at $E = 0$ and *outer corner modes* to the remaining corner states that move away from zero energy.

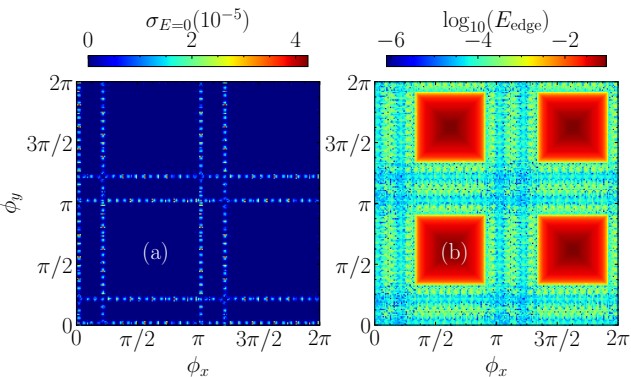

FIG. 9. (a) Standard deviation ($\sigma_{E=0}$) of the energies of the corner modes as a function of the phase shifts ($\phi_x, \phi_y$). (b) Logarithm of edge state energy ($\log_{10}(E_{edge})$) as a function of the phase shifts ($\phi_x, \phi_y$). Results obtained for $W = 3.8$ and $\gamma = 0.2$ in the QPQI phase for a system size $L = 377$.

We now focus on boundary effects on the QPQI phases.

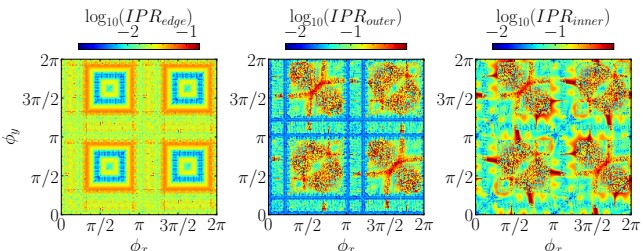

FIG. 10. IPR of the edge states, outer and inner corner modes, plotted from left to right respectively, for $L = 377$.

A distinct feature of QPQI phases is the much stronger dependence on phase shifts. In Fig. 9 we show $\sigma_{E=0}$ and $E_{edge} \to 0$ for the choice of parameters $\gamma = 0.2$ and $W = 3.8$, well inside the QPQI phase. Just as before, regions with $\sigma_{E=0} > 0$ [Fig. 9(a)] are correlated to regions where $E_{edge} \to 0$ [Fig. 9(b)]. Furthermore, increasing system size confines these regions to singular points (for smaller system sizes see Appendix D). This behavior suggests that all the quantities of relevance ($P$, $P_{occ}$ and $q_{xy}$) quantize to their respective values as these regions collapse to singular points, since points have no statistical importance.

As $E_{edge} \to 0$, we expect the localization length of the corner modes to increase as the states hybridize. In Fig. 10 we plot the IPR of the edge states, outer and inner corner modes for $L = 377$. As discussed previously, only the outer corner modes hybridize, as can be seen by the clear pattern emerging in the middle plot of Fig. 10, corresponding to the IPR of the outer corner modes. As the edge states cross the Fermi energy ($E_{edge} \to 0$), the IPR of the outer corner modes decreases, indicating that the localization length increases. Although IPR alone is not enough to conclude about the localization length of a state, for fully delocalized states IPR $\approx \frac{1}{N}$, thus for the considered system size ($L = 377$) we expect IPR $\approx 10^{-5}$. Therefore, an IPR of the order of $10^{-2}$ is a good indication that the corner modes remain localized throughout the ($\phi_x, \phi_y$) plane.

Overall, QPQI phases exhibit strong correlation between the localization length of the corner modes and the edge gap (which is $\phi$ dependent). As the edge states energy approach the Fermi level, they localize (as opposed to the general behavior where edge states are delocalized along the edge) moving towards the corners and hybridizing with the corner modes. This hybridization "pulls" the outer corner modes away from zero energy while increasing their localization length. In the QI regime, the corner modes display oscillating localization length with varying phase shifts, however to a far lesser degree than in the QPQI phases. Furthermore, unlike the QPQI regime, there is no apparent correlation between the localization length of the corner modes and the energy of the edge states, suggesting that the oscillations on the localization

length have a different origin than corner-edge hybridization.

## IV. CONCLUSIONS

We have studied the phase diagram of a quasiperiodic quadrupole chiral insulator. The higher-order topological phases of the parent model were found to be stable upon the addition of quasiperiodicity. More interestingly, quasiperiodicity was also found to induce multiple reentrant topological transitions into quadrupole insulating phases, expanding the known class of chiral symmetric quadrupole insulators introduced in Ref. [79, 88]

Similarly to disorder-induced transitions into higher-order topological Anderson insulators [18, 77, 78], quasiperiodic-induced topological transitions occurring under the usual gap closing and reopening mechanism were observed. However, in contrast to the disordered case, the eigenstates in the gap edge were found to be ballistic across the transition, as in the homogeneous limit. Also in contrast with the disordered case, multiple quasiperiodic-induced topological transitions were found.

A further interesting question that deserves further attention concerns the interplay between the quasiperiodicity and higher-order topology. Quasiperiodicity alone can induce a rich edge physics, which was not the focus of our study. However, these edge states can have a nontrivial interplay with the zero energy corner modes that deserves further exploration.

Our findings can be observed experimentally in different tunable platforms, including electric circuits [13, 89], mechanical metamaterials [12] and photonics [90].

## ACKNOWLEDGEMENTS

The authors acknowledge partial support from FCT-Portugal through Grant No. UIDB/04650/2020. M. G. acknowledges partial support from Fundação para a Ciência e Tecnologia (FCT-Portugal) through Grant No. UID/CTM/04540/2019. M. G. acknowledges further support from FCT-Portugal through the Grant No. SFRH/BD/145152/2019. We finally acknowledge the Tianhe-2JK cluster at the Beijing Computational Science Research Center (CSRC) and the OBLIVION supercomputer through Projects No. 2022.15834.CPCA.A1 and No. 2022.15910.CPCA.A1 (based at the High Performance Computing Center—University of Évora) funded by the ENGAGE SKA Research Infrastructure (Reference No. POCI-01-0145-FEDER-022217—COMPETE 2020 and the Foundation for Science and Technology, Portugal) and by the BigData@UE project (Reference No. ALT20-03- 0246-FEDER-000033—FEDER) and the Alentejo 2020 Regional Operational Program. Computer assistance was provided by CSRC and the OBLIVION support team.

## Appendix A: Edge and Corner States via DOS

Another quantity that is of use is the ratio of DOS defined as $\rho_{OBC}/\rho_{PBC}$. In bulk regions $\rho_{OBC}/\rho_{PBC} \to 1$, however in gapped regions where corner and edge states may live it is expected that $\rho_{OBC}/\rho_{PBC} \gg 1$. This quantity is of particular interest since we can use it to look for topological corner modes at zero energy. In this manner, in a topological phase, a peak of $\rho_{OBC}/\rho_{PBC}$ at zero energy should be observed. In Fig. 12 and Fig. 13 we plot this quantity as a function of $E$ for several $W$'s in each of the observed phases in the two different constant $\gamma$ cuts.

## Appendix B: Constant $W$ Cut

In the main text we have focused our analysis in two constant $\gamma$ cuts. However, a constant $W$ cut was also performed as seen in Fig. 1. We plot several quantities as a function of hopping strength ($\gamma$), such as the quadrupole moment and boundary invariant (a), the bulk spectral gap (b), fractal dimensions (c-d), corner occupation probability (e) and the corner charge (f).

The most interesting result is regarding phase $I$ which is a gapless QPQI phase. Although this regime is gapless, the states at Fermi level are localized which allows for the existence of localized topological corner modes at zero energy. Despite the topological nature of this regime, $P_{occ}$ does not quantize since numerical exact diagonalization of the system Hamiltonian yield linear combinations of localized corner modes and localized bulk states that live at zero energy. Phase $II$ displays several trivial gapped regimes with either a localized or critical gap edge. From phase $II$ to $III$ the gap closes in a ballistic gap-edge into a gapped QI phase with well quantized $q_{xy}$ and $P$. Regarding the corner occupation probability, it plateaus for most of phase $III$, however at around $\gamma \approx 0.5$ it displays two dips that are associated with corner edge hybridization. Furthermore if we consider the constant $\gamma = 0.5$ cut and set $W = 3$ (end of phase $I$, Fig. 2(a)), a small dip can also be observed in the $P$ invariant, in $P_{occ}$ and in $\bar{Q}$. Phase $IV$ is a band gap trivial insulator.

## Appendix C: Phase Twists and Spectral Gap

Introducing phase twists leads to shifts in the allowed $\mathbf{k}$'s, such that if a twist $\theta_i$ is introduced then $\mathbf{k}_i = \left(-\frac{\pi}{L} + \frac{2\pi}{L}n\right)\mathbf{b}_i$. We can add a twist that shifts all the allowed $k_x$'s in such a way that for odd $L$, $k_n = 0$ is a possible value of crystalline momenta. This is achieved by considering $\theta_x = \pi$ for odd $L$ while for even $L$ we use

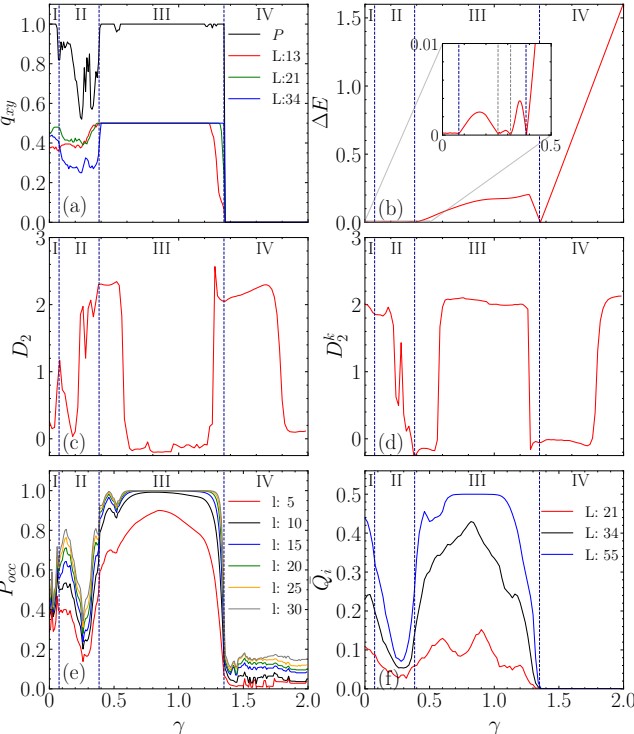

FIG. 11. (a) $q_{xy}$ and $P$ as a function of $W$. The $P$ invariant was computed for a system size of $L = 233$ with 100 averages over phase shifts. $q_{xy}$ was obtained via real space methods with 100 averages over phase shifts. For even $L$, $\theta_x = \theta_y = 0$; for odd $L$ $\theta_x = \theta_y = 0$. (b) Spectral gap as a function of $W$ computed for an even system size $L = 144 \implies \theta_i = 0$ and averaged over 50 phase shifts realizations. (c-d) Fractal dimension averaged out over 100 phase twists, shifts and over the first 8 eigenstates with energies closest to $E = 0$. (e) Corner occupation probability ($P_{occ}$) as a function of $W$ for the first 2 zero energy states. The results were obtained for a system size of $L = 377$ and averaged over 100 phase shifts. (f) Corner charge ($\bar{Q}$) as a function of $W$ for several system sizes and averaged over 100 phase shifts and over the 4 corners. (e-f) l $\delta = 10^{-5}$.

regular periodic boundary conditions which is the same as considering twists with $\theta_x = 0$. This trick is also relevant for gap dependent results as topological phases and TPT. A particular choice of twists is also relevant in some disordered systems, although it is not clear that in the absence of translational symmetry, using phase twists is advantageous. In fact, even for some disordered system, phase twists solve spectral gap finite size effects. Quasiperiodic systems in the low $W$ regime maintain ballistic states at Fermi level and so setting a non zero twist can improve gap calculations, especially when the gap closes in a ballistic-ballistic transition. In contrast regular uncorrelated disorder does not benefit from phase twists, since all of the electronic states are Anderson localized (for low dimensional systems).

## Appendix D: Phase Shift Dependence

In Fig. 14 we expand on the results of Fig. 10 by plotting the standard deviation ($\sigma_{E=0}$) and logarithm of edge state energy ($\log_{10}(E_{edge})$) as a function of the phase shifts ($\phi_x, \phi_y$) for smaller system sizes.

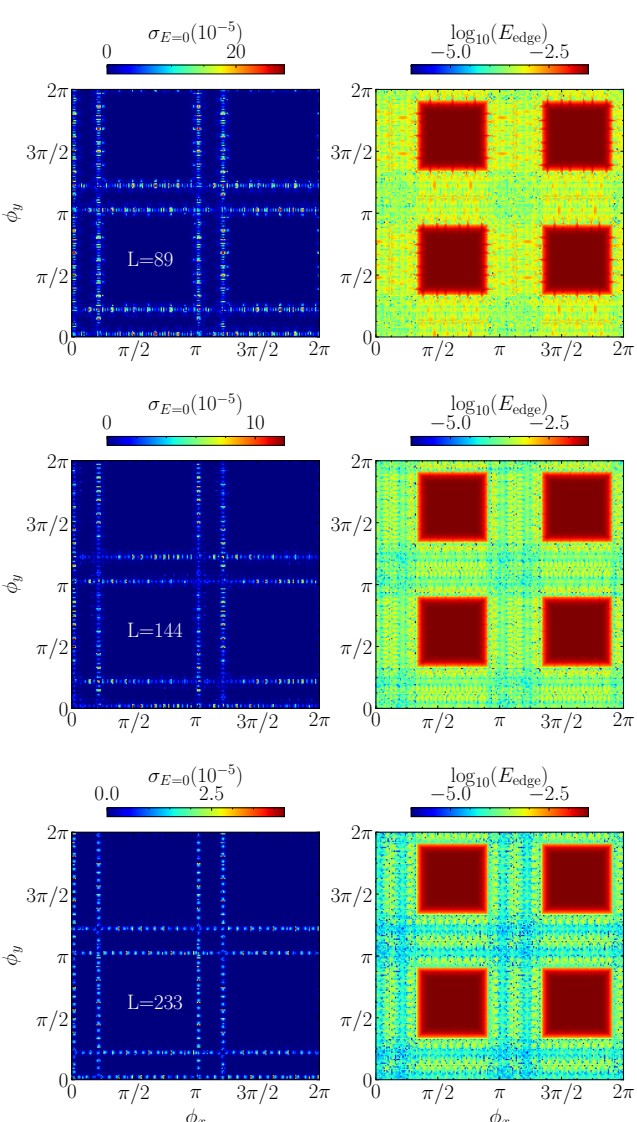

FIG. 14. Standard deviation ($\sigma_{E=0}$) of the energies of the corner modes as a function of the phase shifts ($\phi_x, \phi_y$) and logarithm of edge state energy ($\log_{10}(E_{edge})$) as a function of the phase shifts ($\phi_x, \phi_y$). Results obtained for $W = 3.8$ and $\gamma = 0.2$ in the QPQI phase for a system sizes $L = \{89, 144, 233\}$ as indicated.

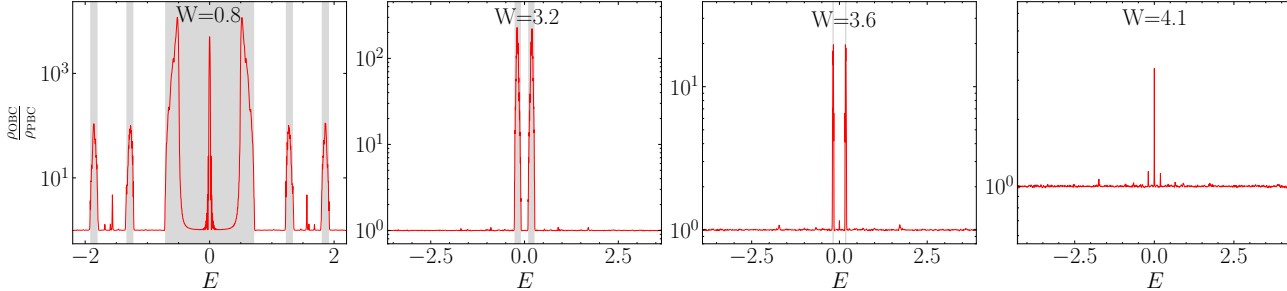

FIG. 12. Ratio of open boundary DOS ($\rho_{OBC}$) and periodic boundaries DOS ($\rho_{PBC}$) as a function of energy for $W$ in different phases and $\gamma = 0.5$. The greyed out regions correspond to bulk gaps opened by the QP modulation. The following KPM parameters were used: $L = 987$, $M = 5000$, $R = 10$.

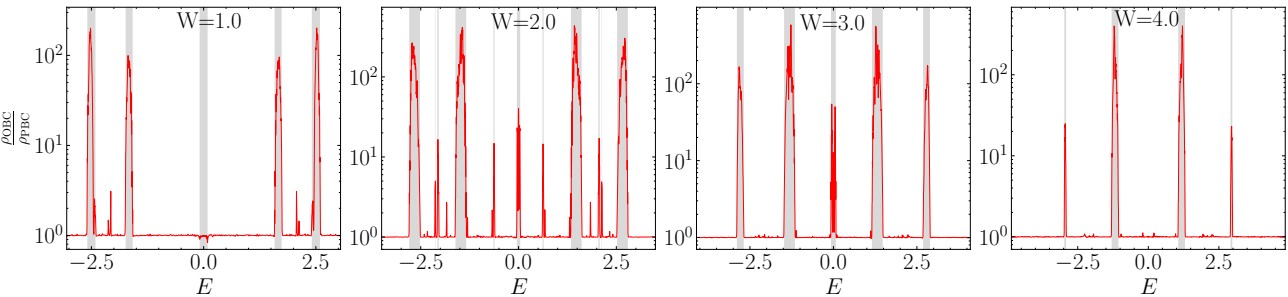

FIG. 13. Ratio of open boundary DOS ($\rho_{OBC}$) and periodic boundaries DOS ($\rho_{PBC}$) as a function of energy for $W$ in different phases and $\gamma = 1.1$. The greyed out regions correspond to bulk gaps opened by the QP modulation. The following KPM parameters were used: $L = 987$, $M = 5000$, $R = 10$.

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
