# Peer review of "Quasiperiodic Quadrupole Insulators"

_SciPost Physics_

## Round 2 · Referee Report · Anonymous (Referee 1) · 2024-10-2

Report

This manuscript does not meet the journal's criteria number 1 and 2:

  • Provide a novel and synergetic link between different research areas (they provide a synergetic link, but it is not novel)

  • Open a new pathway in an existing or a new research direction, with clear potential for multi-pronged follow-up work (no new pathway/already existing results)

Attachment

Recommendation

Ask for major revision

  • validity: -
  • significance: -
  • originality: -
  • clarity: -
  • formatting: -
  • grammar: -

Author:  Raul Liquito  on 2025-01-30  [id 5166]

(in reply to Report 1 on 2024-10-02)

Warnings issued while processing user-supplied markup:

  • Inconsistency: plain/Markdown and reStructuredText syntaxes are mixed. Markdown will be used.
    Add "#coerce:reST" or "#coerce:plain" as the first line of your text to force reStructuredText or no markup.
    You may also contact the helpdesk if the formatting is incorrect and you are unable to edit your text.

Response to referee 1 report Raul Liquito, Miguel Gonçalves, Eduardo V. Castro October 14, 2024

Dear Referee 1,

First of all we thank the referee for this speedy report. In this document we address the comments made the referee one by one. At the end of the document we provide a list of changes made in the paper.

Kind regards,

The authors.

Note: In the attachments I added a pdf file with this response.

============================================================================

Referee: The paper does not open a new research idea, but rather extends the availability of topological phases already existing with the very broad literature. Moreover, a very similar research paper was uploaded in arXiv before this one: 2406.13535 (now published as PRB 110, 075422). They have the same overarching conclusion: the clean phase diagram is enlarged by the presence of the quasiperiodic modulation. This paper is not cited by the authors. Additionally, the new region found in the current paper was not found in the previous one. This might be because of the slightly different quasiperiodic function chosen in this paper, but this has to be clearly explained and a thorough comparison with the current work must be done.

Authors: We thank the referee for calling our attention to the research paper available at arXiv 2406.13535 (now published as PRB 110, 075422). That paper was announced on 21 June 2024, followed by our work (arXiv 2406.17602) announced five days later (26 June 2024) on the same platform. Although the first referee report was made available in October, to no fault of the referee, our paper was submitted to Scipost three months earlier, one week after its submission on arXiv. The papers were submitted to PRB and SciPost, respectively, with a difference of 8 days between each other. Most importantly, the two works are independent from each other and, in our opinion, our work has relevant, non-overlapping findings with the paper now published as PRB 110, 075422. For these reasons, we believe that supplementing our work with a thorough comparison with PRB 110, 075422 is out of the scope of the submitted manuscript. However, we are open to add a final note to the article: “After completing this work, we became aware of a related work [cite].”.

Furthermore, we firmly believe that there are significant differences between our work and PRB 110, 075422. In particular, the QPQI phase, one of the main results in our work, has not been previously reported in the literature. In contrast with the QI and T phases that are adiabatically connected to the limit of vanishing quasiperiodicity, the QPQI can only be induced by a gap reopening at strong quasiperiodicity, showing a rich interplay between higher-order topology and quasi-periodic topology inherited from higher dimensions, as we detail in the manuscript. Our work showed the first example, to our knowledge, where both kinds of topology can be intertwined in the same phase. This, in our opinion, paves the way for exploring the nature and properties of this new quasiperiodic quadrupole insulating phase.

Referee: There are many other works exploring how quasiperiodicity affects a higher order quadrupole insulator. For example, 1904.09932 or 2001.07551. A quick online search reveals that there are several more published with similar research questions and results. A more thorough literature review must be conducted by the authors.

Authors: We thank the referee for the comment. Following the referee suggestion, we performed a more extensive literature review and introduced some references in the introduction. There are some works exploring higher order topology in non crystalline systems, including in quasicrystals [Phys. Rev. Lett. 124, 036803][Phys. Rev. B 102, 241102(R)][Phys. Rev. Lett. 123, 196401], amorphous lattices [Phys. Rev. Research 2, 012067(R), or hyperbolic lattices [Phys. Rev. B 107, 184201]. In our work, we focus on a different type of quasiperiodic system (non quasicrystal), which more correctly should be referred to as incommensurate quasiperiodic systems. Quasicrystals, by definition, have rotational symmetries that are not allowed in periodic systems. In fact, higher order phases protected by these symmetries can be realised in quasicrystals with eight-fold [Phys. Rev. Lett. 124, 036803][Phys. Rev. Lett. 123, 196401], twelve-fold [Phys. Rev. B 102, 241102(R) ], and five-fold [Nano Letters 2021 21 (16), 7056-7062] rotational symmetries. Our work tackles a different type of quasiperiodic system where all crystalline symmetries of the system are broken, including the rotational symmetries, and where higher-order topology has not been studied yet, except for the new work PRB 110, 075422.

Referee: Some acronyms have never been introduced, such as KPM and DOS (p.2)

Authors: We thank the referee for pointing this out. We introduced both acronyms the first time they are mentioned.

Referee: The Hamiltonian in Eq.1 introduces ΨR, but then writes that is composed of $c_{k\alpha}$. This is probably a typo. Moreover, it contains a matrix $\Lambda_y$, which is not Hermitian.

Authors: We thank the referee for pointing these typos to us. We corrected $c_{k\alpha}$ to $c_{R\alpha}$. The matrix $\Lambda_y$ is well defined, but the Hamiltonian is missing its hermitian conjugate (h.c.) on the hopping term, which we have corrected.

Referee: The authors state that the φ-dependence should disappear in the thermodynamic limit. They should either prove this statement or give a reference where it is proven.

Authors: We address this comment further ahead.

Referee: The notation for the Bloch Hamiltonian of the clean BBH model is not ideal. The $\Gamma$ matrices overlap with the previous $\Gamma_R$ matrix.

Authors: We agree with the referee and changed the notation accordingly: we replaced $\Gamma_R$ with $\Delta_R$.

Referee: The matrix $\Gamma_R$ , in Eq.2, contains a term proportional to $\delta$, which was never defined.

Authors: In the text we introduced the definition of $\delta$. “…and $\delta$ is a staggered mass term in each sublattice which opens a gap proportional to $\delta$ …”

Referee: The operator $\hat{n}(R)$ below Eq.6 was never defined. Also, it should be with lowercase $r$.

Authors: We thank the referee for pointing the typo and for the suggestion. We introduced the proper definition of the operator $\hat{n}(R)$ below eq. 6. Since $\hat{n}(R)=\sum_\alpha c_{R\alpha}^\dagger c_{R\alpha}$, it follows that it should be written with uppercase $R$, since the operator is defined inside the Hilbert space spanned by $c_{R\alpha}^\dagger$.

Referee: Below Eq.8, there is a typo with $\hat{p}_i$, which should be $\hat{p}i(r)=\frac{x_i\hat{n}(r)}{L$}.

Authors: We corrected the expression below eq. 8 to $\hat{p}i(R)=\frac{x_i\hat{n}(R)}{L$}. We note that capital $R$ should be used since the operator $\hat{n}(R)$ is defined inside the Hilbert space spanned by $c_{R\alpha}^\dagger$.

Referee: The authors should provide more details on the derivation of the boundary Green’s function in Eq.7.

Authors: We thank the referee for the suggestion. We have made modifications to the paragraph between eq. 6 and 7, to better describe the method. We also cite the appropriate paper where the method is explained.

Referee: The authors do not explain why they average over both phase twists θ and phase shifts φ. They should also explain why they introduce twisted boundary conditions in the main text, and not only in the appendix, or at least refer to the appendix.

Authors: Here we also address “The authors state that the φ-dependence should disappear in the thermodynamic limit. They should either prove this statement or give a reference where it is proven.”

Generally, phase twists impact the energy spectra when the corresponding eigenfunctions exhibit real space delocalization. The opposite occurs with phase shifts, affecting only the energy spectra when the eigenfunctions are localized or critical [see “papers do Miguel” SciPost (arXiv:2103.03895), PRB (arXiv:2206.13549), PRL (arXiv:2208.07886)]. Nonetheless, in either case, the localization properties of the eigenstates are not dependent on the choice of phase shifts and twists. Thus, we average over both quantities to ensure smoother results. Regarding phase twists, in finite systems only specific ks are allowed in the FBZ. Introducing a phase twist shifts all the allowed crystalline momenta, and an improper choice of the twists can originate a spectral gap. For this reason, when calculating spectral gap-sensitive results, such as topological phase transitions, we fix the twists to 0 or pi to ensure that the spectral gap of the system is converged for all system sizes. In the thermodynamic limit, all ks are allowed within the FBZ, and thus, phase twists will not affect the spectra of the system.

As for the phase shifts, we can understand why the φ-dependence should vanish for the bulk properties in the thermodynamic limit in the following simple case. For a quasiperiodic potential in a square lattice, the phase shifts appear as twists in the reciprocal space hamiltonian. Thus phase shifts are transformed to boundary conditions in reciprocal space and have no impact in the thermodynamic limit. More generally, in an infinite quasiperiodic system, all local configurations of the quasiperiodic potential are realised, and thus introducing a phase leads to no change in the bulk states. In finite systems, a simple way of probing all the possible configurations present in the infinite system, is to average over different phase shifts, a common practice in the literature, see e.g. [Phys. Rev. Lett. 120, 207604,ANNALEN DER PHYSIK 2017, 529, 1600399, Phys. Rev. B 104, L041106]. In contrast, edge properties, such as the edge spectrum, retain a dependence on phase shifts even in the thermodynamic limit. The reason is that different shifts change the local configurations at the boundaries, no matter the size of the system. The Aubry-André model is the paradigmatic example where this phenomenon can be observed, where the edge states disperse with the phase shift, as discussed e.g. in [PRL 109, 106402].

We rewrote the section where phase shifts and twists are discussed, further explaining why averages are taken over these quantities. We added a new reference [Phys. Rev. B 108, L100201] where it is shown that the energy spectrum of the system is dependent on the choice of phase shifts and twists. We refer to the appendix for a more detailed explanation of how a finite spectral gap can appear in a finite system with an appropriate choice of twists.

Referee: In Fig.2, they repeat (a) and (b) twice.

Authors: We thank the referee for the correction, and have made the proper changes.

Referee: The authors have never defined what is $IPR_k$.

Authors: We introduced the definition of $IPR_k$ below eq. 10.

Referee: The authors have not described how the the normalized localization length ξ/M is calculated from the transfer matrix matrix approach. They should clearly state its definition.

Authors: Added a description of the transfer matrix method, clarifying how the normalised localisation length is calculated. Also added references to the articles where TMM is introduced.

Referee: In Fig.2(b), region II is subpartitioned into a,b and c. There seems to be a very tiny gap with ∆E <10−3, which they claim results in the peak that goes toward Qxy = 0.5 and P = 1. This signal is extremely weak, as they mention themselves. Moreover, since the localization lengths are inversely proportional to this gap size, the zero modes will always hybridize and be pushed away from E=0 in the OBC system.

Authors: We thank the referee for the comment. We agree that due to the small gap size, the zero modes could hybridize and be pushed away from E=0 in the OBC system. However, there are several instances of topological metals and semimetals exhibiting bound states in the continuum. In the present case, for particular choices of phase shifts, the quadrupole moment is quantized to 0.5, while for other it quantizes to 0.0. Indeed, for these particular phase shifts, the zero energy corner modes hybridize with critical gap-edge bulk states, but this behavior is not generic for all phase shifts. Such a phase shift sensitivity is similar to the one observed in the QPQI phase. Since the gap is very small, this hybridisation phenomena is stronger, occurring for more phase shift pairs. However, our results show that the gap seen in IIIb is robust with increasing system size. Moreover, the peak seen in the P invariant scales to 1 as the system size is increased.

Referee: The authors, claim that there are multiple reentrant transitions. From the phase diagram, one can only see one such transition. The previously mentioned region II does not offer sufficient evidence to consider it a proper transition. Moreover, it is typically the case that the very small gaps happen because of finite-size effects and disappear in the thermodynamic limit.

Authors: We thank the referee for this comment. Although region II is a very fined grained region, the fact that the gap scales to a finite value and the P invariant scales to 1 with linear system size up to L = 610 is, in our opinion, very strong evidence that this gap is robust in the thermodynamic limit. Moreover, the existence of corner modes at zero energy for some phase shifts, while for other the outer corner modes hybridize with the critical gap-edge states, fully agrees with the conclusion that phase IIIb is a QPQI phase. For this reason, we believe our claim about multiple reentrant transitions is accurate.

Referee: The QPQI phase, which is the main novelty of this work, also suffers from a particular shortcoming: the zero-energy edge modes easily hybridize with edge states when the latter are close to E = 0, pulling away the zero modes and increasing their localization lengths. This breaks their degeneracy and one wonders how resilient the quantization of the quadrupole is against these effects.

Authors: We thank the referee for the comment. The presence of edge states, which hybridize with the corner modes for certain values of $(\phi_x,\phi_y)$, is a distinctive property of the QPQI phase. In a finite system, this hybridization occurs only when the edge states energy $E_{edge}$ is smaller than the mean level spacing of the system. Therefore, in the thermodynamic limit this phenomenon will only be observed for special values of $(\phi_x,\phi_y)$ when $E_{edge}=0$. The quadrupole moment will thus be quantized to its topological value except for the particular phase shifts where corner modes hybridize with edge states. As seen in Fig. 15 (left column), the regions in $(\phi_x,\phi_y)$ plane where the hybridization takes place are shrinking, suggesting they will be of zero measure in the thermodynamic limit. On the other hand, for finite systems the phase shifts are tunable physical parameters, and therefore the edge modes can be tuned to be either away or close in energy to corner modes, varying their hybridization. This is an interesting new property of the QPQI phase, rather than a shortcoming.

Attachment:

reply_referee_1_final.pdf

---

## Editorial Decision

resubmitted